# Morphologic diversity of cutaneous sensory afferents revealed by genetically directed sparse labeling

Hao Wu[1], John Williams[1,4], Jeremy Nathans[1,2,3,4]*

[1]Department of Molecular Biology and Genetics Johns Hopkins University School of Medicine, Baltimore, United States; [2]Department of Neuroscience Johns Hopkins University School of Medicine, Baltimore, United States; [3]Department of Ophthalmology Johns Hopkins University School of Medicine, Baltimore, United States; [4]Howard Hughes Medical Institute, Johns Hopkins University School of Medicine, Baltimore, United States

**Abstract** The diversity of cutaneous sensory afferents has been studied by many investigators using behavioral, physiologic, molecular, and genetic approaches. Largely missing, thus far, is an analysis of the complete morphologies of individual afferent arbors. Here we present a survey of cutaneous sensory arbor morphologies in hairy skin of the mouse using genetically-directed sparse labeling with a sensory neuron-specific alkaline phosphatase reporter. Quantitative analyses of 719 arbors, among which 77 were fully reconstructed, reveal 10 morphologically distinct types. Among the two types with the largest arbors, one contacts ~200 hair follicles with circumferential endings and a second is characterized by a densely ramifying arbor with one to several thousand branches and a total axon length between one-half and one meter. These observations constrain models of receptive field size and structure among cutaneous sensory neurons, and they raise intriguing questions regarding the cellular and developmental mechanisms responsible for this morphological diversity.

## Introduction

The skin is the largest organ that senses external stimuli, and the responses of sensory afferents in the skin encompass a diversity of modalities richer than those any other sensory organ. Among these modalities are thermosensation, pain, itch, and many distinct types of mechanosensation. A further level of complexity arises from the distinctive anatomic distributions of the different types of cutaneous sensory afferents. For example, hairy skin has specialized mechanosensors that contact hair follicles and their associated Merkel cell clusters, whereas glabrous (i.e. palmar) skin is innervated by specialized mechanosensors, such as those that contact Meissner corpuscles, that are absent from hairy skin. Additionally, the spatial density of sensory afferents and the spatial scale of sensory integration vary widely over the body surface. For example, two-point discrimination for mechanical stimuli has high resolution on the fingertips and low resolution on the torso (*Iggo, 1982*), and sensory thresholds for hot and cold stimuli vary in a fine-grained mosaic over the body surface (*Norrsell et al., 1999*).

Early physiologic insights into cutaneous afferent diversity came from the discovery that different types of receptors are associated with distinct conduction velocities and pharmacologic thresholds for conduction block (*Erlanger et al., 1924*; *Gasser and Erlanger, 1927*, *1929*). Recordings from peripheral nerves also revealed differences in stimulus threshold and in adaptation to sustained stimuli, especially among different subtypes of mechanoreceptors (*Iggo, 1982*). Structural insights have come from light and electron microscopic studies that have defined numerous specialized nerve endings and their associated structures and correlated them with different cutaneous sensory modalities. These

*For correspondence: jnathans@jhmi.edu

**eLife digest** Sensory neurons carry information from sensory cells in the eyes, ears and other sensory organs to the brain and spinal cord so that they can coordinate the body's response to its environment and various stimuli. The sensory organs responsible for four of the traditional senses—vision, hearing, smell and taste—are relatively small and self-contained: however, the sensory organ responsible for touch is as big as the body itself. Moreover, a variety of many different types of sensory cells in the skin allow the body to respond to temperature, pain, itches and a range of other external stimuli.

Despite more than a century of research, relatively little is known about the morphology of the complex networks (arbors) of sensory neurons that send signals towards the central nervous system. This is mainly due to difficulties involved in imaging intact skin, the way that different arbors overlap and intermingle, and the relatively large distances that separate the bodies of neuronal cells and the farthest reaches of their arbors.

Wu et al. employed an imaging method that exploits the Cre-Lox system that is already widely used in genetics. In this approach a Cre enzyme is used to remove a region of DNA that is flanked by two genetically engineered Lox sequences. Wu et al. used a gene that codes for an enzyme marker (alkaline phosphatase) that previous investigators had inserted into the DNA of mice. The gene was inserted in such a way that it was only expressed in sensory neurons that innervate the skin when Cre-Lox recombination had removed an adjacent segment of DNA. Moreover, Wu et al. used this reporter gene in combination with a modified Cre enzyme that only enters the nuclei of cells in the presence of a drug (Tamoxifen), so the probability that the marker gene is expressed is determined by the concentration of Tamoxifen. By administering a low level of Tamoxifen to pregnant mice, it was possible to label a very small number of sensory neurons in each embryo. Individual neurons that express the alkaline phosphatase marker were visualized with a histochemical reaction that rendered them dark purple. The remainder of the tissue remained unstained.

Based on quantitative analyses of the morphologies of more than 700 arbors, Wu et al. identified 10 distinct types of neurons. Of the two types of neurons with the largest arbors, one makes contact with ~200 hair follicles, with the nerve endings completely encircling the follicles; the other type of arbor contains several thousand branches, with a total length for all of the branches summing to as much as one meter in length. The next challenge is to study the morphologies of neurons in tissues other than the skin, and also the neurons involved in other sensory systems, and to explore the cellular and developmental mechanisms responsible for the morphological diversity found in these initial experiments.

structures include free endings, Pacinian and Meissner corpuscles, Raffini endings, Kraus end bulbs, Merkel cell clusters, and lanceolate and circumferential endings associated with hair follicles (*Iggo and Andres, 1982*).

Over the past 15 years the diversity of cutaneous receptor subtypes has been placed on a foundation that is increasingly defined in molecular terms. The identification of receptor proteins—such as TRP channels for pain and temperature, Mrg receptors for itch, and various receptors for inflammatory compounds—and their localization to subsets of DRG neurons has provided the most direct molecular classification of sensory neurons (*Basbaum et al., 2009*; *Liu et al., 2009*). Cutaneous receptor subtypes also express distinctive sets of transcription factors, neuropeptides, neurotrophin receptors, and miscellaneous cytosolic proteins, which can be used in combination to provide an empirical classification (*Fundin et al., 1997*; *Marmigère and Ernfors, 2007*; *Reed-Geaghan and Maricich, 2011*).

Largely missing thus far from the field of cutaneous sensory biology is an analysis of the full morphologies of sensory afferents. While the cross-sectional microanatomy of cutaneous sensory afferents has been intensively studied, their full arbors have been technically difficult to visualize for two reasons. First, the analysis of complete arbor morphologies requires either single cell tracer filling or genetically-directed sparse expression of a reporter, followed by imaging of full thickness skin to visualize individual labeled arbors against a background in which the vast majority of arbors are unlabeled. With respect to tracer filling methods, since cutaneous arbors generally reside at a substantial distance from their cell bodies in the DRG, tracer diffusion from the cell body is inefficient. Second, because

skin is a relatively refractile tissue, immunostaining and imaging of full-thickness skin preparations has not been widely practiced. This second challenge has largely been solved with the recent development of protocols that give high signal-to-noise ratio immunostaining and organic solvents that render intact skin optically clear (*Li et al., 2011*). In contrast to the challenges associated with visualizing sensory projections in the skin, a number of investigators have analyzed the projections of individual dorsal root ganglion (DRG) neurons in the dorsal laminae of the spinal cord and have correlated these projection patterns with molecular and physiologic properties (e.g. *Woodbury et al., 2001*).

Defining the full morphologies of cutaneous sensory afferents is of interest for many reasons. Most obviously, it would inform our understanding of receptive field size and structure. It could also reveal patterns of target innervation and territoriality that could be used to classify sensory neurons and complement classifications based on physiologic and molecular criteria (*Marmigère and Ernfors, 2007*; *Reed-Geaghan and Maricich, 2011*). In other systems, such as the vertebrate retina, the integration of morphologic and physiologic information has provided deep insights into structure, function, and evolution (*Rodieck and Brening, 1983*; *Masland, 2001, 2011*).

In the present study, we have used genetically-directed sparse labeling in adult mouse back skin to survey many hundreds of cutaneous afferent arbors and to generate a collection of complete arbor morphologies. Quantitative parametric analysis reveals 10 discrete classes within this collection, several of which have arbors of extraordinary length and complexity.

## Results

### Genetically-directed sparse labeling of cutaneous sensory afferents

Brn3a, a POU-domain transcription factor, is expressed in the vast majority of DRG and trigeminal ganglion neurons beginning at ~E11 (*Fedtsova and Turner, 1995*). Neurons expressing *Brn3a* can be visualized using a *Brn3a* conditional knockout allele (*Brn3a$^{CKOAP}$*) that has one *loxP* site in the 5' untranslated region (UTR), a second *loxP* site 3' of the 3' UTR, and an alkaline phosphatase (AP) reporter distal to the second *loxP* site (*Badea et al., 2009*). Cre-mediated excision of the *Brn3a* coding region and 3' UTR activates expression of *AP* by placing it under the control of the *Brn3a* promoter. In the present study, sparse Cre-mediated recombination was obtained using a *Neurofilament Light Chain* (*NFL*)-*IRES-CreER* knock-in allele and low dose Tamoxifen (*Rotolo et al., 2008*; see 'Materials and methods'). *NFL-IRES-CreER* was chosen as the source of Cre-recombinase because it is widely expressed in projection neurons, it is not expressed in non-neural tissue, and it produces a relatively low level of CreER. By contrast, the combination of *Brn3a$^{CKOAP}$* with a ubiquitously expressed CreER (*ROSA26-CreER*; *Badea et al., 2003*) activates *AP* expression in muscle and connective tissue as well as in DRG neurons, thereby compromising the clarity with which cutaneous sensory afferents can be imaged. *Brn3a$^{+/−}$* and *Brn3a$^{+/+}$* mice appear to be indistinguishable in appearance and overall health and individual *Brn3a$^{+/−}$* DRG neuronal cell bodies do not differ in appearance or number relative to *Brn3a$^{+/+}$* controls (*Xiang et al., 1996*). Importantly, *Trieu et al. (2003)* and *Eng et al. (2004)* have shown that, in *Brn3a$^{+/−}$* DRG neurons, a Brn3a-dependent negative feedback regulatory system leads to nearly wild type levels of Brn3a transcripts and other Brn3a-regulated transcripts. Thus, it seems unlikely that *Brn3a$^{+/−}$* DRG neurons differ functionally or morphologically from their wild type counterparts.

The present survey of afferent arbor morphologies was conducted with back skin because this territory includes a wide variety of cutaneous sensory types and its large area facilitates the identification of well-isolated AP-stained arbors. In mature pigmented mice, melanin in skin and hair confounds full-thickness skin imaging. This difficulty was circumvented by harvesting the skin at P21, the midpoint of the ~2-day telogen phase of the highly synchronous first hair cycle (*Müller-Röver et al., 2001*; *Alonso and Fuchs, 2006*). During this time window, skin pigmentation is temporarily lost (*Figure 1A*). Titration of the Tamoxifen dose at gestational day (GD)17 showed that for the *Brn3a$^{CKOAP/+}$*; *NFL-IRES-CreER/+* genotype, 200 µg, 500 µg and 1 mg of intraperitoneal (IP) Tamoxifen produced ~5, ~50 and >500 labeled and well isolated arbors per back skin at P21 (*Figure 1B,C*, and *Figure 1—figure supplement 1*). At the highest Tamoxifen dose (1 mg), individual sensory arbors cannot be resolved (*Figure 1—figure supplement 2*).

A total of 101 P21 *Brn3a$^{CKOAP/+}$*; *NFL-IRES-CreER/+* back skins were analyzed following maternal exposure to 100, 200, 250, or 500 µg of Tamoxifen at GD17. With an average surface area of 15.53 cm$^2$ per skin, this corresponds to a total of 1569 cm$^2$ examined for AP stained sensory arbors. The fraction of back skin surface area occupied by well-separated AP+ sensory arbors varied from ~0.2% to ~15%.

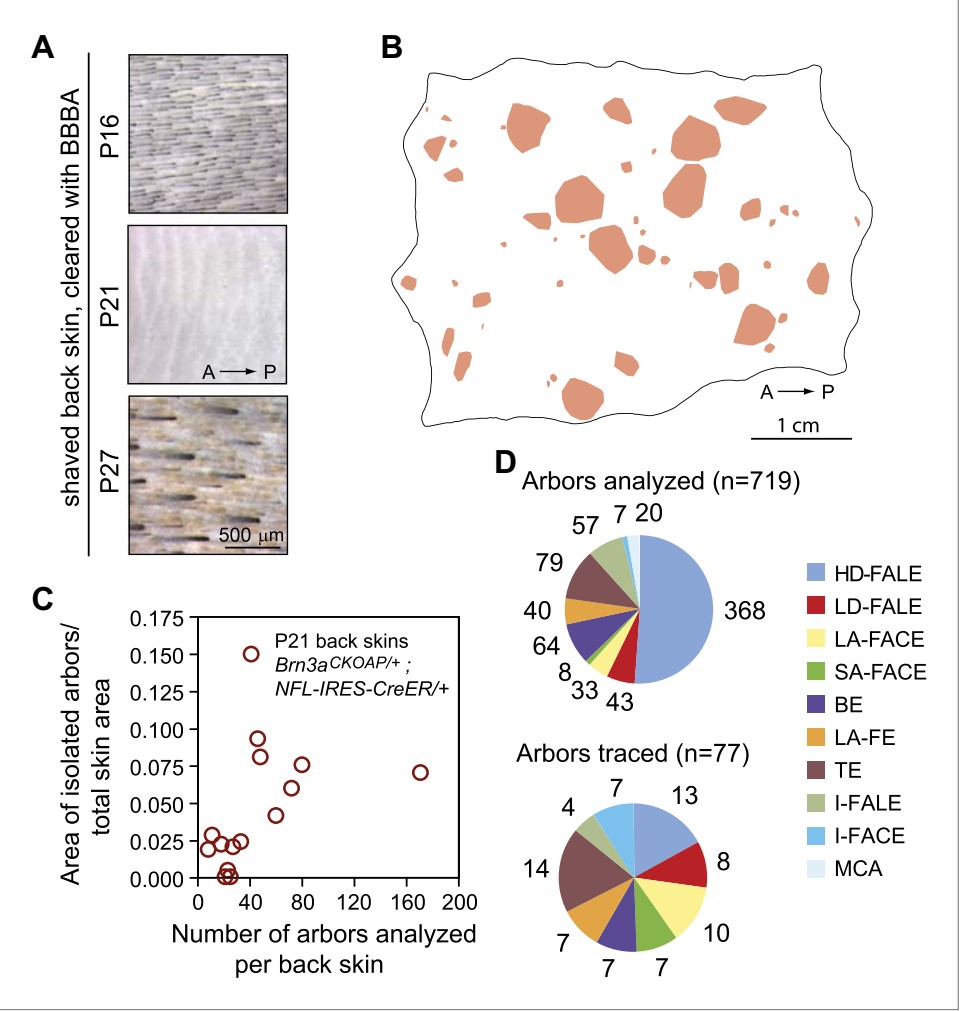

**Figure 1**. Genetically-directed sparse labeling of cutaneous sensory afferents. (**A**) Shaved back skin at P16, P21, and P27 shows the nadir of pigmentation at P21. (**B**) Isolated AP+ arbors that were included in the arbor area survey are represented by convex pink polygons on a P21 *Brn3a*$^{CKOAP/+}$; NFL-IRES-CreER/+ back skin. A, anterior; P, posterior. (**C**) Plot showing the fraction of total skin area occupied by isolated AP+ arbors vs the number of AP+ arbors included in the arbor area survey. (**D**) Number of cutaneous arbors of each type analyzed for arbor area (upper pie chart) and traced (lower pie chart). No MCA arbors were traced.
The following figure supplements are available for figure 1.

**Figure supplement 1**. Eight skins showing well-separated cutaneous sensory afferent territories at P21.
**Figure supplement 2**. P21 skin with a high density of AP+ cutaneous sensory arbors.

A total of 719 arbors that appeared by visual inspection to be free from overlap were characterized further (***Figure 1C,D***). To analyze arbor morphologies in detail, we traced 77 full arbors for nine different arbor types (***Figure 1D***) using Neuromantic, a publicly available reconstruction program.

The set of labeled sensory neurons is predicted to correspond to the intersection of the expression domains of the *Brn3a* and *NFL* genes. Although *Brn3a*$^{CKOAP}$ is expressed in nearly all DRG neurons (***Badea et al., 2012***), expression of the *NFL-IRES-CreER* knock-in allele presumably mirrors the abundance of the neurofilament light chain and is therefore enriched in sensory neurons with large axon diameters. Thus the current survey likely covers only a fraction of the morphologic diversity of cutaneous sensory arbors. We also note that the abundances of different arbor types within this set is not related in any simple way to the actual abundances of these types within the skin because (1) variations in the

level of *NFL-IRES-CreER* expression in different neuronal types will bias their representation, (2) larger arbors tend to be under-represented because the probability of arbor overlap increases with size, and (3) the representation of the most abundant arbor classes was limited by the investigators to a number sufficient for statistically robust analysis.

In the paragraphs that follow we describe the identification and characterization of 10 morphologically distinct cutaneous arbor types. These types were initially distinguished by visual inspection. Subsequent statistical analyses of their morphologic properties (arbor area, axon length, number of axon branches, type of sensory ending specialization, number of hair follicles contacted, and lamination depth within the skin) have confirmed those divisions. At present, the correspondence between several of these morphologic types and previously characterized sensory neuron types cannot be made with certainty, and for this reason we will refer to the 10 types using names that are based strictly on their morphologies. These are (in alphabetical order): bushy ending (BE), high density follicle-associated lanceolate ending (HD-FALE), isolated follicle-associated circumferential ending (I-FACE), isolated follicle-associated lanceolate ending (I-FALE), large area follicle-associated circumferential ending (LA-FACE), large area free ending (LA-FE), low density follicle-associated lanceolate ending (LD-FALE), Merkel cell associated (MCA), small area follicle-associated circumferential ending (SA-FACE), and thick ending (TE). (Here we use 'lanceolate endings' to mean longitudinally oriented lanceolates, i.e. parallel to the follicle axis; transverse lanceolate endings have also been described (*Fundin et al., 1997*).) Individual arbors are referred to by type, followed by an individual identifier in parenthesis, for example I-FALE (A13-11). In the figures that follow, all of the images and arbor traces are viewed in the plane of the skin unless otherwise noted.

## Arbors that contact Merkel cell clusters (MCA) and isolated follicles (I-FACE and I-FALE)

The smallest afferent arbors appear to target single Merkel cell clusters (touch domes) or single hair follicles. In the latter category (I-FACE and I-FALE), most of the follicles appear to be guard (tylotrich) hairs based on their large diameter (~50 µm). The assumption that MCA, I-FACE, and I-FALE afferents target single Merkel clusters or single follicles must be qualified by noting that all cutaneous sensory axons enter the dermis from subdermal nerves and are, therefore, invariably broken during the skin dissection, generally within several hundred microns of their terminal arbors. Since we cannot rule out the possibility that one or more additional branches emerge from the parent axon at a point proximal to its broken end, we refer to the follicle targets as 'isolated' rather than 'single'. This caution is given extra weight by evidence that an individual axon can innervate more than one Merkel cell cluster based on neurobiotin filling experiments in neonatal mice. In these experiments several slowly adapting type I (SAI) low threshold mechanoreceptor axons were observed to innervate two adjacent Merkel cell clusters (*Woodbury and Koerber, 2007*). However, in our P21 skin preparations 20/20 labeled MCA arbors targeted only isolated Merkel cell clusters, with no other labeled MCA arbors seen within a radius of >350 µm.

The top row of images in *Figure 2A* shows two arbors with lanceolate endings that encompass almost the entire circumference [I-FALE (A13-11)] or approximately half of the circumference [I-FALE (A13-12)] of an individual guard hair follicle. Two examples of arbors that innervate all or nearly all Merkel cells within a single semi-circular Merkel cell cluster are seen in the bottom row of images in *Figure 2A*. Like Merkel cell clusters, the MCA terminal arbors encompass the posterior 50–75% of the circumference of the central follicle. *Figure 2B* shows tracings of three I-FALE arbors and four I-FACE arbors, the latter circumferentially wrapping three isolated guard hair follicles and one smaller follicle.

## Arbors with bushy endings (BE) and thick endings (TE)

A distinct class of arbors, which we refer to as bushy ending (BE), have dense and finely branched processes that encompass an area of 0.5–1 mm in diameter and ramify at a depth of 5–10 µm from the skin surface (*Figure 3*). As with all arbor types described here, the terminal branches of the BE arbors ramify within a narrow stratum (*Figure 3B*; top image). (The analysis of arbor depth within the dermis is noted at various points in the 'Results' section and is presented quantitatively in *Figure 8D*.) BE arbors have total axon lengths of ~10 cm with ~1,000 branch points (*Figure 3F*). Additional BE arbors, including a comparison of two independent traces of the same BE arbor are shown in *Figure 3—figure supplements 1–3*.

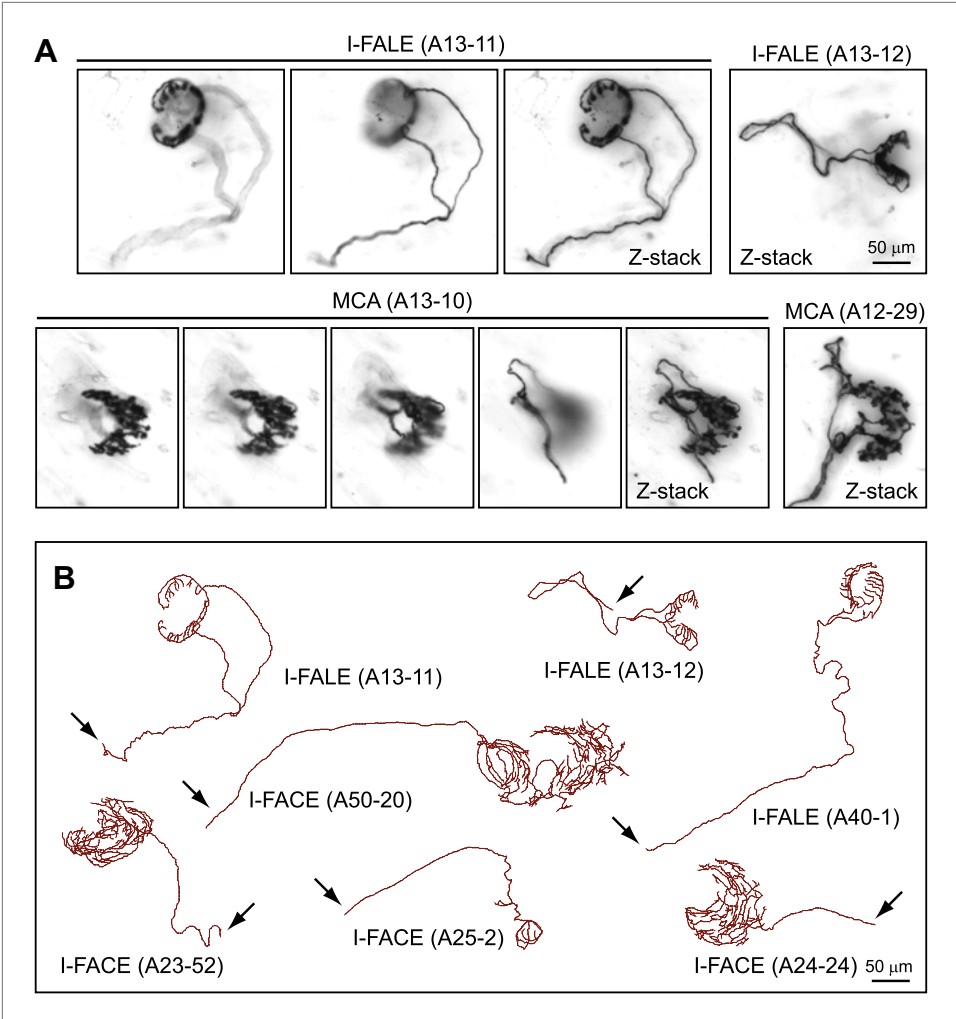

**Figure 2**. Arbors that innervate Merkel cell clusters (MCA) and isolated follicles (I-FACE and I-FALE). (**A**) Upper panels, two arbors that consist of lanceolate endings surrounding an isolated follicle (I-FALE). Shown for I-FALE (A13-11) are two Z-planes with superficial (left) and deep (right) layers, with a Z-stacked image. Lower panels, two arbors that innervate an isolated Merkel cell cluster (MCA). Shown for MCA (A13-10) are four Z-planes from superficial (left) to deep (right) layers, with a Z-stacked image. (**B**) Tracings of three I-FALE arbors and four I-FACE arbors. Arrows indicate the afferent axon.

Another distinctive type of arbor, which we refer to as thick ending (TE), has only 20–80 branch points and exhibits relatively sparse coverage of an irregular area of 300–500 µm in diameter at a depth of ~20 µm from the skin surface (*Figure 4G–J*). As its name implies, a distinctive feature of the TE class is a thickening of the terminal 50–100 µm of its branches, a feature that is presumably related to its sensory function (*Figure 4G*).

## Arbors that contact multiple follicles with lanceolate endings (LD-FALE and HD-FALE)

Two types of arbors that innervate multiple follicles with lanceolate endings were observed. One type, low-density follicle-associated lanceolate endings (LD-FALE), has a sparsely branched arbor that innervates between 2 and 36 follicles (*Figure 4A–D*). These arbors encompass territories of 1–2 mm in diameter, but with total axon lengths of only 5–35 mm and with only 5–40 branch points (*Figure 4I*). (A quantitative analysis of the number of follicles innervated by this and several other arbor types are presented in *Figure 8C*.) Among LD-FALE arbors, some branches terminate with thickened endings that lack the

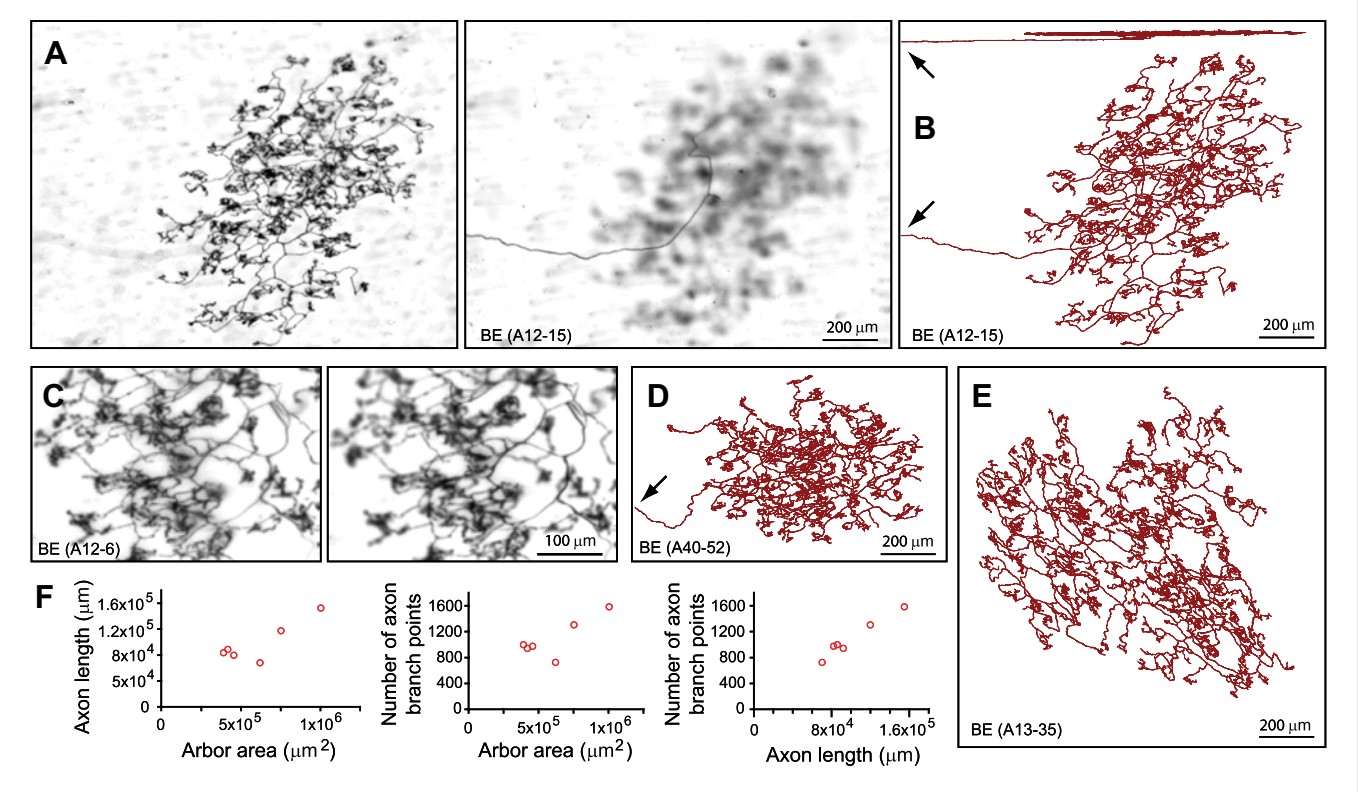

**Figure 3**. Arbors with bushy endings (BE). (**A**) Arbor BE (A12-15), showing Z-planes with superficial (left) and deep (right) layers. The single axon that gives rise to this arbor is seen at the left in the deep layer image. (**B**) Tracing of BE (A12-15) in side view (top) and en face (bottom). Arrows in (**B**) and (**D**) indicate the afferent axon. (**C**) Higher magnification images of a portion of BE (A12-6) at two Z-planes. (**D**),(**E**) Tracings of BE (A13-35) and BE (A40-52). (**F**) Relationships between arbor area, number of axon branch points, and axon length for six traced BE arbors.

The following figure supplements are available for figure 3.

**Figure supplement 1**. Additional BE arbor tracing.

**Figure supplement 2**. Additional BE arbor tracing.

**Figure supplement 3**. Additional BE arbor tracing.

full C-shape of the typical follicle-associated lanceolate endings; these may represent endings that contact only a small fraction of a follicle's circumference or that are in the process of growing around or retracting from a target follicle. In nearly every instance, the opening of the 'C' faces the anterior of the mouse (*Figure 8E*).

The most commonly encountered arbor type, high density follicle-associated lanceolate ending (HD-FALE), accounts for ~50% of all AP+ arbors (*Figure 1D*). The lanceolate endings of HD-FALE arbors localize ~50 µm beneath the skin surface, and each arbor contacts ~90% of the follicles within an area that ranges from ~200 µm to ~1 mm in diameter (*Figure 5*). Although most HD-FALE arbors have areas less than $10^6$ µm² (mean ± SD of $3.4 ± 3.2 × 10^5$ µm²) and innervate fewer than 50 follicles (mean ± SD of 30 ± 22), this type varies substantially, with arbor areas ranging from $10^5$ µm² to $2.5 × 10^6$ µm² and the number of follicles innervated ranging from 6 to 153 (*Figure 5D*). For the 12 traced arbors, the mean axon length (excluding the lanceolate endings) was ~$2 × 10^4$ µm and the mean number of branch points was ~75. As seen with I-FALE and LD-FALE arbors, individual HD-FALE endings wrap around their target follicles to variable extents (*Figure 5A–C*), covering the full circumference or as little as ~30% of the circumference.

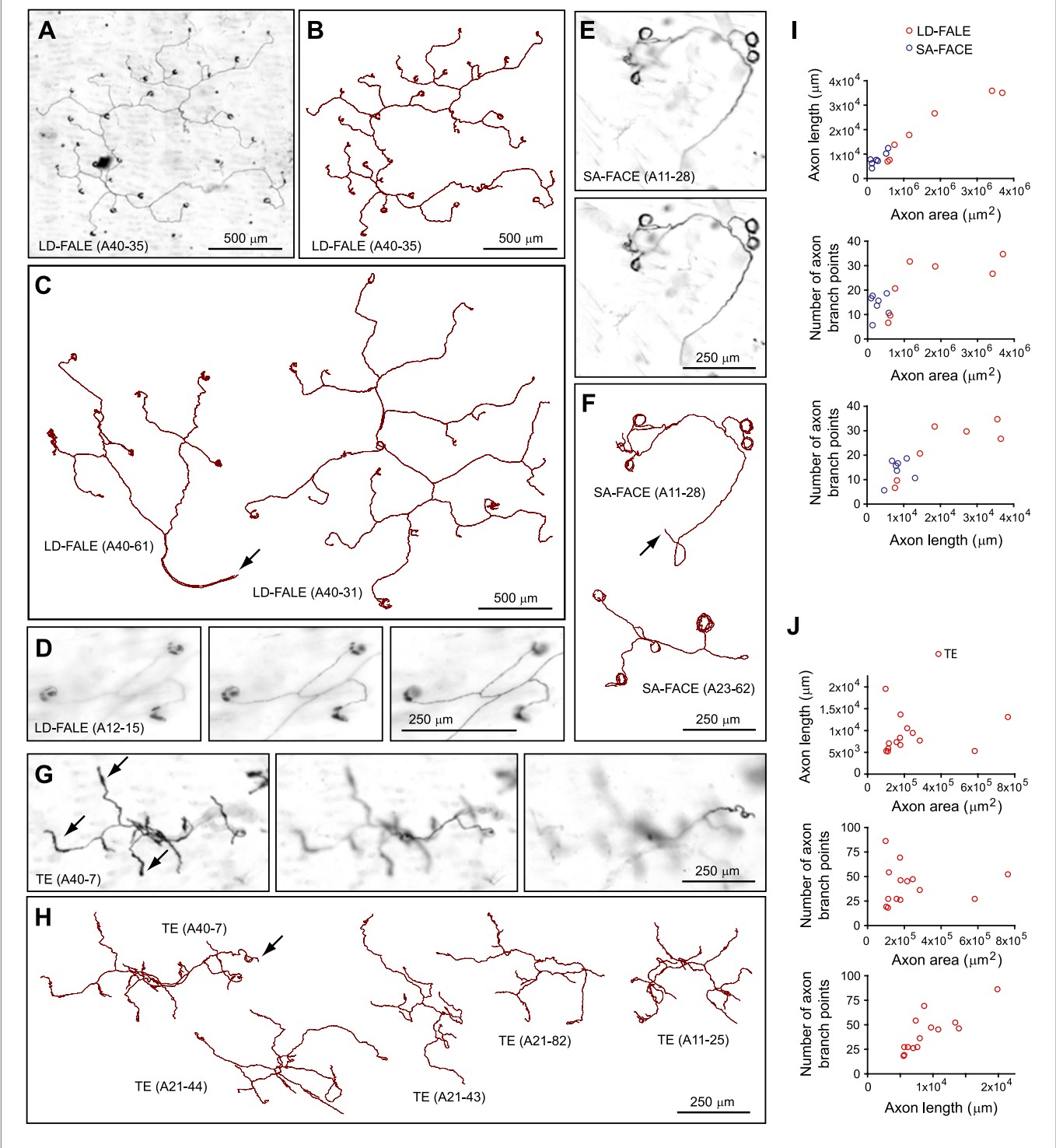

**Figure 4**. Arbors with low density follicle-associated lanceolate endings (LD-FALE), small area follicle-associated circumferential endings (SA-FACE), and thick endings (TE). (**A**),(**B**) LD-FALE (A40-35) image (**A**) and trace (**B**). (**C**) Tracings of LD-FALE (A40-31) and (A40-61). Arrows in (**C**), (**F**), and (**H**) indicate the afferent axon. (**D**) Higher magnification images at three Z-planes (left to right, superficial to deep) of LD-FALE (A12-15) showing three lanceoalate endings. (**E**) SA-FACE (A11-28), showing Z-planes with superficial (top) and deep (bottom) layers. (**F**) Tracings of SA-FACE (A11-28) and SA-FACE (A23-62). (**G**) TE (A40-7), showing three Z-planes from superficial (left) to deep (right) layers. Arrows indicate the thickened nerve terminals of the TE arbors. (**H**) Five TE tracings. TE (A40-7), shown in panel G, is at the upper left. (**I**),(**J**) Relationships between arbor area, number of axon branch points, and axon length for 7 SA-FACE, 7 LD-FALE, and 14 TE traced arbors.

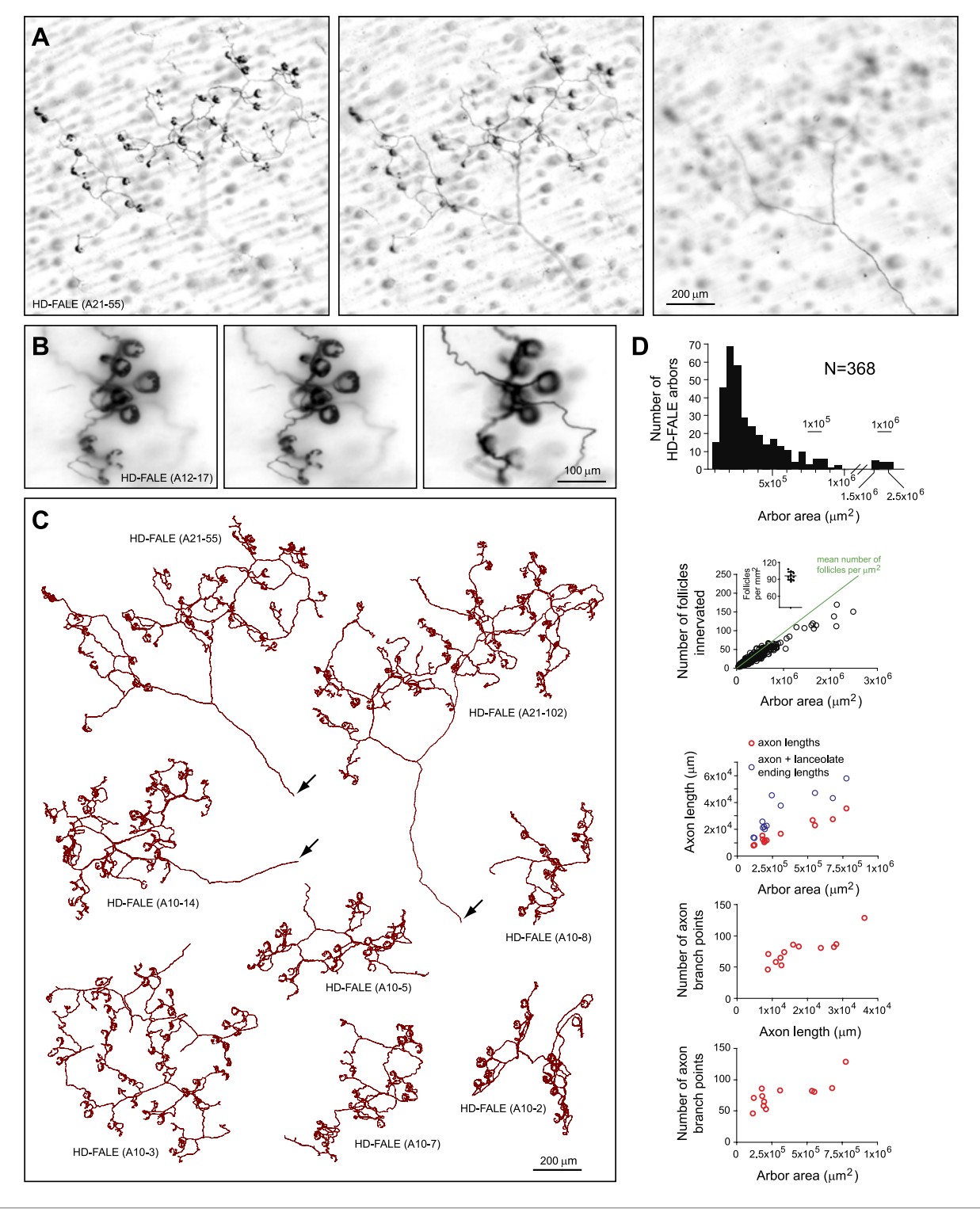

**Figure 5**. Arbors with high density follicle-associated lanceolate endings (HD-FALE). (**A**) Arbor HD-FALE (A21-55) showing superficial (left), intermediate (center), and deep (right) layers. The single axon that gives rise to this arbor is seen at the lower right of the deep arbor image. (**B**) Higher magnification images at three Z-planes (left to right, superficial to deep) of HD-FALE (A12-17) showing multiple lanceolate endings. (**C**) Tracings of eight HD-FALE arbors. HD-FALE (A21-55), shown in panel A, is in the upper left. Arrows indicate the afferent axon. (**D**) For all HD-FALE arbor areas analyzed, arbor area and number of follicles innervated per arbor are shown in the top two plots. In the second plot, the inset shows the determination of the number of follicles per µm²; this value is shown in the plot as a green line. The bottom three plots show the relationships between arbor area, number of axon branch points, and axon length for the 12 traced HD-FALE arbors.

## Arbors that contact multiple follicles with circumferential endings (SA-FACE and LA-FACE)

Two types of arbors that innervate multiple follicles with circumferential endings were observed. One type, which we refer to as small area follicle-associated circumferential ending (SA-FACE) was rarely observed (8 out of 719 arbors). SA-FACE follicles contact few follicles (mean ± SD of 3.8 ± 1.1), generally have fewer than 20 branch points, and have diameters of ~200 to ~500 μm (*Figure 4E,F,I*). By contrast, arbors designated as large area follicle-associated circumferential endings (LA-FACE) contact many follicles (mean ± SD of 197 ± 73), including occasional guard hair follicles, within territories of 1–2 mm in diameter (*Figure 6*). These arbors have axon lengths (excluding the circular endings) that cluster in the 8–15 cm range, with ~200 branch points per arbor. The fraction of follicles innervated within the arbor territory ranges from 19% to 79% with a mean of 57%. There is a clear trend for larger arbors to exhibit a lower mean density of innervated follicles (*Figure 6E*).

## Large arbors with free endings (LA-FE)

The largest arbors appear to comprise a single morphologic class, referred to as large arbors with free endings (LA-FE; *Figure 7*, *Figure 7—figure supplements 1 and 2*). In this class, the axons are arranged in two narrowly stratified tiers: an inner arbor at a depth of ~70 μm and an outer arbor at a depth of ~10 μm from the skin surface are connected by >50 vertical branches (*Figures 7A,B and 8D*). The inner arbor exhibits sparse but relatively uniform coverage of an area ~3 mm in diameter with a total axon length of ~10–15 cm. The outer arbor exhibits denser and relatively uniform coverage of an area ~4 mm in diameter with a total axon length of 50–100 cm (*Figure 7B–D*, *Figure 7—figure supplements 1 and 2*). By way of comparison, we note that the torso of a P21 mouse is only ~4 cm in length. The axons appear to lack terminal specializations. The relatively uniform coverage within the outer arbor is not achieved by a strict self-avoidance mechanism, as there are numerous examples of axon crossings within the same arbor. However, within the inner arbor, axon crossings are rare.

## Parametric analyses

In the descriptions above, clear qualitative distinctions between most arbor types are apparent based on the presence or absence of follicle-associated endings, the type of specialized structure elaborated by the sensory endings, and whether or not the arbor targeted single guard hair follicles or Merkel clusters. The quantitative analysis of continuous variables in *Figure 8* reinforces those distinctions and, in addition, provides a clear separation between the two pairs of arbor types that are most closely related by qualitative criteria, LD-FALE vs HD-FALE and SA-FACE vs LA-FACE.

*Figure 8A* compares arbor areas on both linear and $\log_{10}$ scales. This analysis reveals different degrees of clustering of area values within different arbor types, with BE, LA-FACE, and LA-FE types showing the tightest clustering. By contrast, variation in arbor areas within HD-FALE, LD-FALE, and TE types spans two orders of magnitude. *Figure 8A* also shows that the areas of SA-FACE and LA-FACE arbors are non-overlapping. Extending this analysis to the correlation of axon length and arbor area illustrates striking differences in arbor density, with the high density BE arbors showing the highest length:area ratio and the sparse LD-FALE arbors showing the lowest ratio (*Figure 8B*). This plot also shows a clear separation between LA-FACE and LA-FE arbor types and between these and other types.

For those arbors that contact hair follicles, the number and density of follicle contacts dictate the scale of spatial integration and the sampling density within the relevant area. *Figure 8C* compares these parameters for each of the four arbor types that contact multiple follicles. In this figure we introduce a new parameter, the 'innervation index', defined as the number of follicles innervated per μm² of skin. It is apparent that the two classes of arbors with circular endings, LA-FACE and SA-FACE, differ markedly in number of follicles innervated: >100 per LA-FACE arbor and <10 per SA-FACE arbor (*Figure 8C*, left panels). SA-FACE arbors also have a lower mean innervation index than LA-FACE, and in a plot of innervation index vs arbor area the two distributions are non-overlapping (*Figure 8C*, right panels). The two classes of arbors with lanceolate endings, HD-FALE and LD-FALE are not clearly distinguished by the number of follicles innervated (*Figure 8C*, left panels), but show a clear separation in a plot of innervation index vs arbor area (*Figure 8C*, right panels).

Skin is a highly stratified tissue in which individual structures such as arrector pili muscles, sebaceous glands, and Merkel cells occupy distinct strata. *Figures 8D* and *Figure 8—figure supplement 1* show

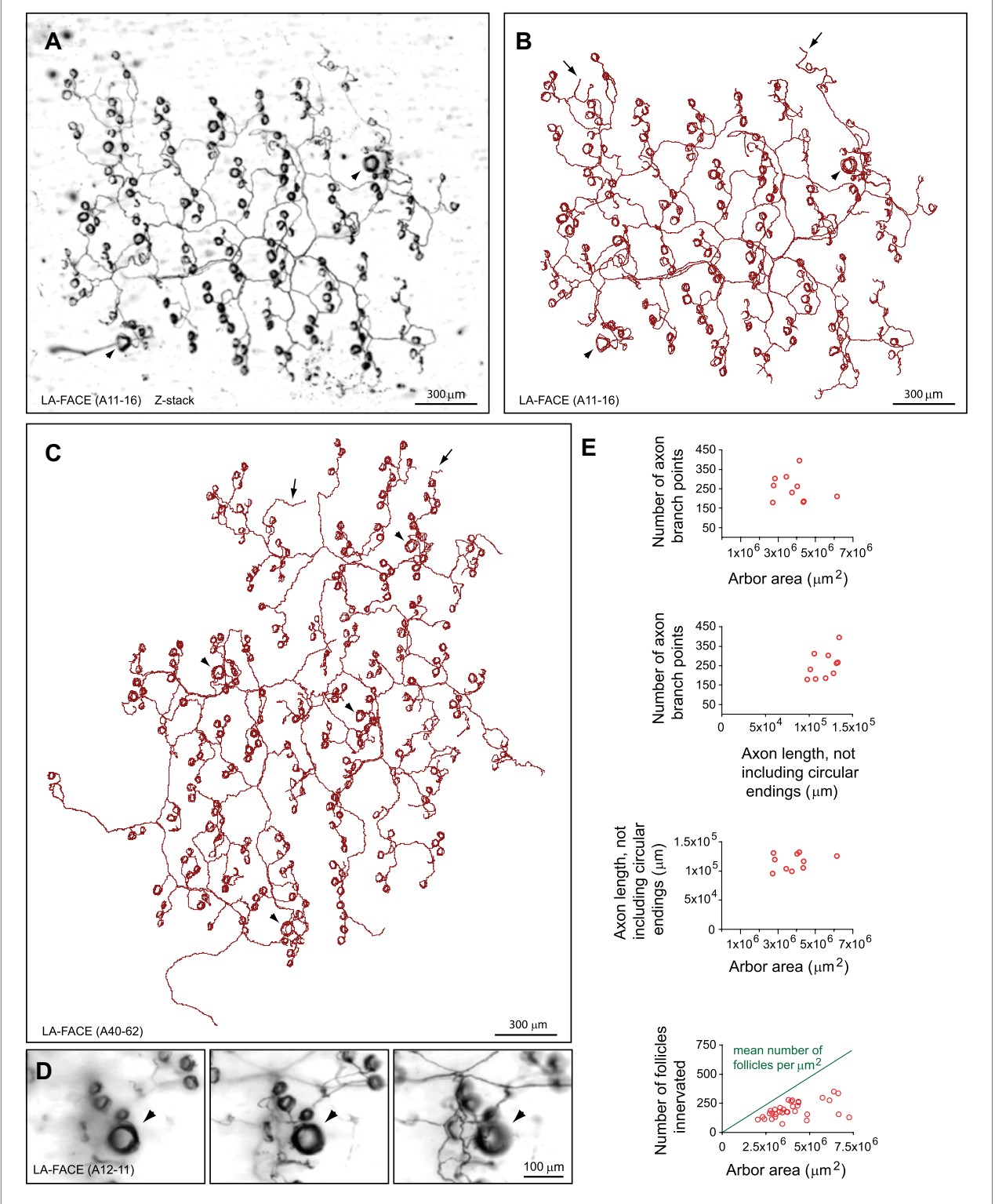

**Figure 6**. Arbors with large area follicle-associated circumferential endings (LA-FACE). (**A**),(**B**) LA-FACE (A11-16) image (**A**) and trace (**B**). Arrowheads in panels B–D indicate guard hair follicles. Arrows in panels B and C indicate occasional nerve endings not associated with follicles. (**C**) Tracing of LA-FACE (A40-62). (**D**) High magnification images at three Z-planes (left to right, superficial to deep) of LA-FACE (A12-11). (**E**) Top, relationships between arbor area, number of axon branch points, and axon length for 10 traced LA-FACE arbors. Bottom, number of follicles innervated per arbor vs. arbor area for 34 LA-FACE arbors. The mean number of follicles per unit area of back skin is shown by the green line (see **Figure 5D** inset).

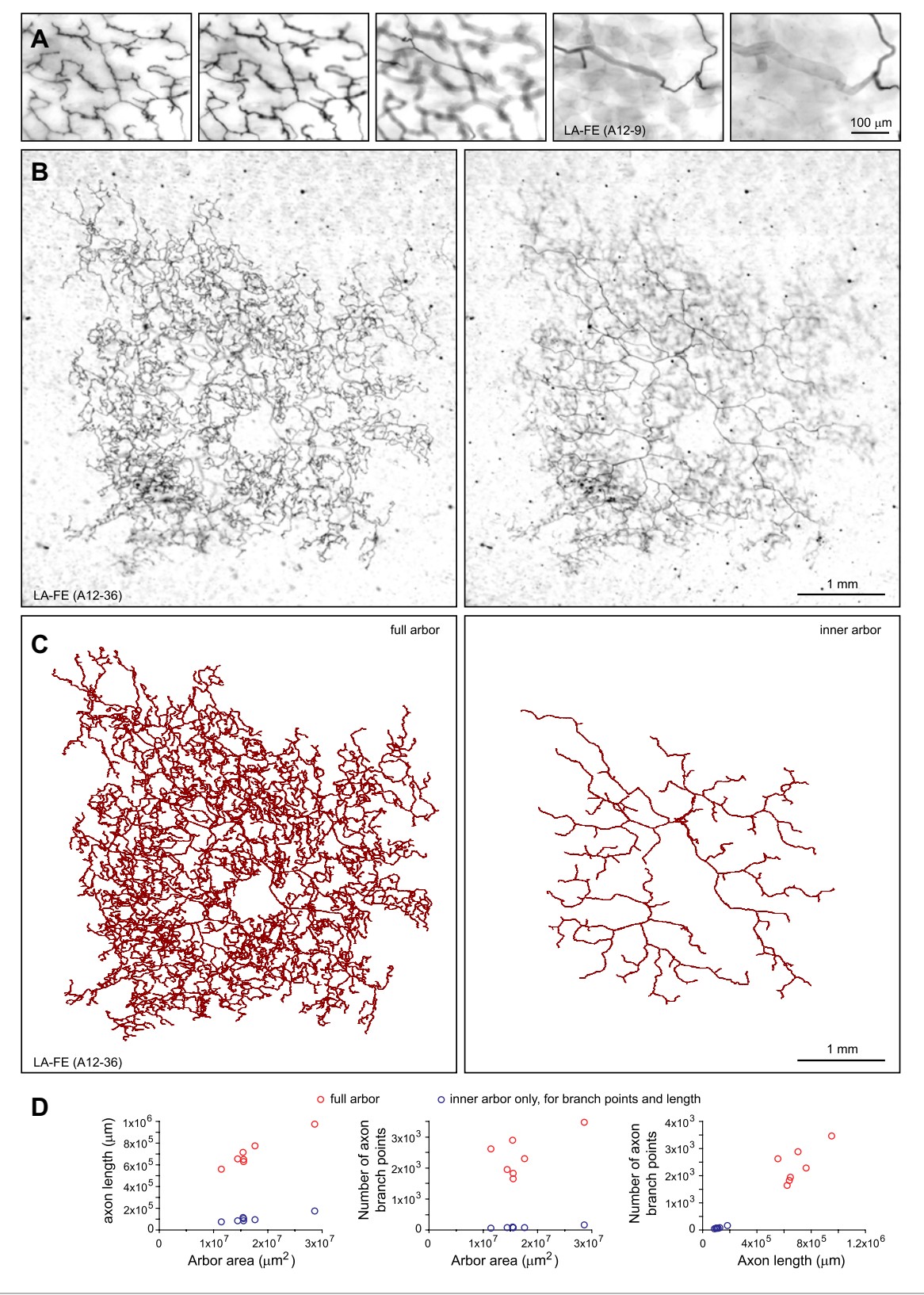

**Figure 7**. Arbors with large areas and free endings (LA-FE). (**A**) High magnification images at five Z-planes (left to right, superficial to deep) of LA-FE (A12-9). (**B**) Arbor LA-FE (A12-36), showing Z-planes with superficial (left) and inner (right) layers. A distinct inner arbor is in focus in the right panel. (**C**) Tracings of the full arbor of LA-FE (A12-36) (left), and its inner arbor alone (right). (**D**) Relationships between arbor area, number of axon branch points,
*Figure 7. Continued on next page*

*Figure 7. Continued*

and axon length for seven traced LA-FACE arbors, with separate analyses for the inner arbor and full arbor. Note that the axon length scale extends to 1 m.

The following figure supplements are available for figure 7.

**Figure supplement 1**. Additional LA-FE arbor tracing.

**Figure supplement 2**. Additional LA-FE arbor tracing.

that cutaneous sensory arbors are precisely stratified and that each arbor type has a characteristic depth. The outer arbors of the BE and LA-FE classes reside close to the skin surface, most likely within the epidermis, and their inner arbors reside ~70 µm from the skin surface. TE arbors are located ~20 µm from the skin surface. Follicle associated circular and lanceolate endings reside ~50 µm from the skin surface for non-guard hair follicles and ~40 µm from the skin surface for guard hair follicles.

## Discussion

The work reported here is the first systematic description of sensory afferent morphologies in verte-brate skin. The 10 arbor types characterized from mouse back skin cover a wide range of morphologies but likely provide a still incomplete picture of cutaneous arbor diversity for at least two reasons. First, the AP reporter and CreER driver used here are expressed in many but not all DRG neurons (*Badea et al., 2012*); and, second, body surfaces such as the lips, cornea, nipples, genitalia, and glabrous skin were not included in the survey. The division of afferent arbors into 10 types is based on a combination of discrete and continuous parameters. The discrete parameters correspond to the morphologies of the endings, which are either lanceolate, circumferential, free, bushy, thick, or Merkel cell-associated. The continuous parameters correspond to the number of follicles innervated, axon length, number of branch points, arbor area, and arbor depth. Visual inspection of the distributions of these parameters produces an unambiguous division into the 10 arbor types defined here, and therefore we have not used more sophisticated methods such as unsupervised clustering, which are typically applied to datasets with less obvious divisions (e.g., see *Badea and Nathans, 2004*).

One important result from this survey is that each sensory arbor exhibits a high degree of uniformity in its terminal structures. Thus, arbors with lanceolate endings never produce circular endings, bushy endings, or thick endings; arbors with circular endings never produce lanceolate endings, bushy endings, or thick endings; and so on. The only possible exception to this pattern is the presence of occasional free nerve endings within arbors that have follicle-associated circular endings (LA-FACE arbors; *Figure 6B,C*). However, these are readily distinguished from the endings of arbors that exclu-sively produce free endings (LA-FE arbors) by virtue of their failure to branch and their deeper stratifi-cation within the dermis. As our analysis was conducted at a relatively young age (P21), these rare free endings could represent growing or regressing branches in a still-plastic arbor. The morphologic uni-formity of terminal structures—presumably the sites of transduction of the sensory stimulus—within each arbor type argues for a corresponding uniformity in the sensory transduction apparatus and, consequently, in the range and types of favored stimuli. Thus, these data support the view that cutaneous sensory afferents represent a set of labeled lines, collectively delivering to the dorsal spinal cord at least 10 parallel representations of external stimuli that impinge on hairy skin.

### Cutaneous sensory arbors and the classification of sensory cell types

For many of the arbor types described here, it is possible to suggest a likely correspondence with physiologically or histologically defined classes of cutaneous sensory neurons (*Fundin et al., 1997*; *Woodbury and Koerber, 2007*; *Li et al., 2011*). MCA arbors likely correspond to the A-beta slowly adapting type I/low threshold mechanoreceptors (SA I/SA LTMRs) that contact Merkel cell clusters/touch domes. I-FALE arbors that surround guard hair follicles likely correspond to A-beta rapidly adapting (RA) LTMRs. HD-FALE arbors likely correspond to A-beta RA LTMRs that innervate awl/auchene hair follicles. I-FACE, LA-FACE, and SA-FACE arbors likely correspond to the peptidergic and/or nonpeptidergic circumferential free nerve endings described by *Fundin et al. (1997)*; the physiologic

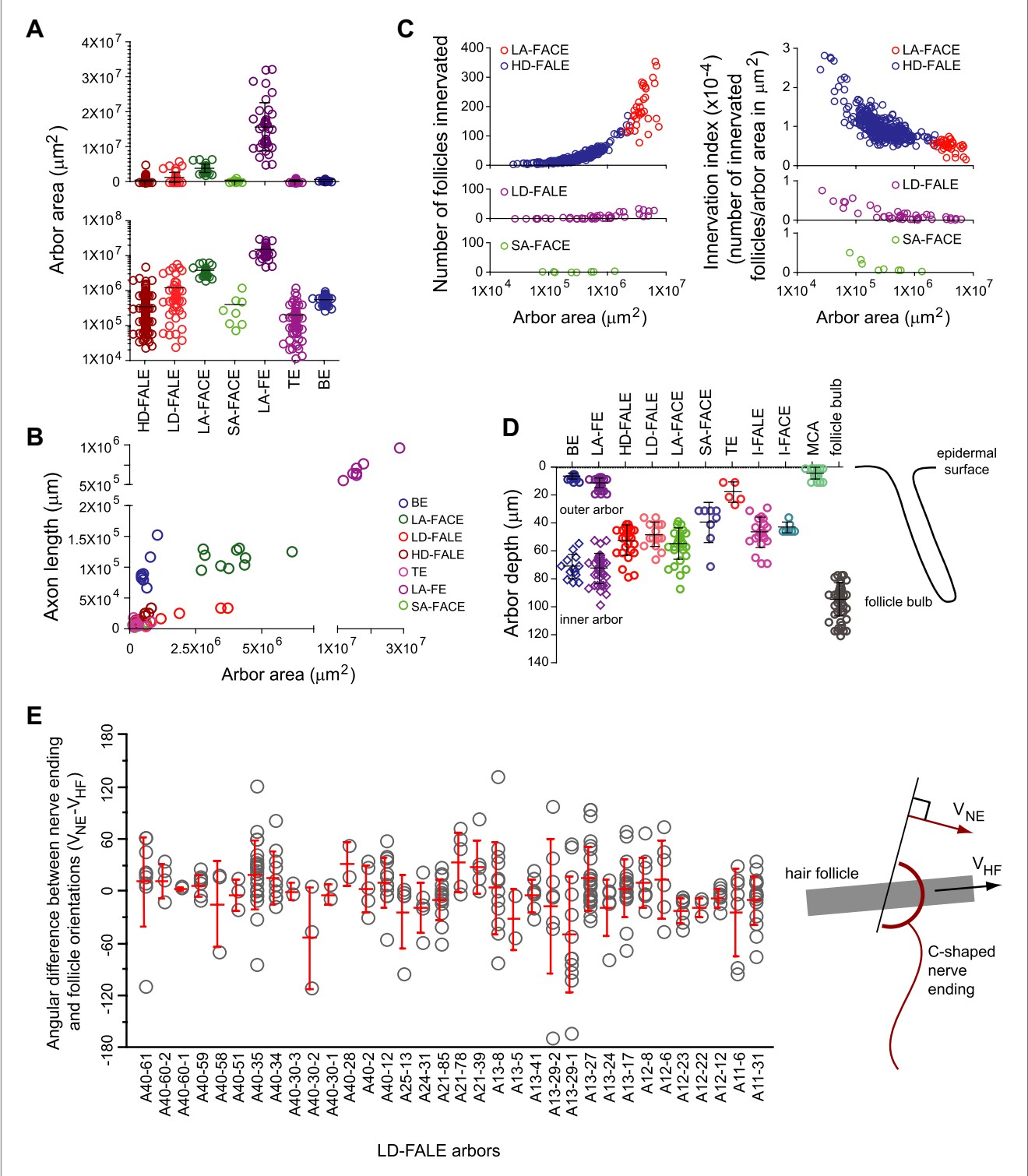

**Figure 8**. Parametric analysis of arbor area, arbor depth, axon length, and number of follicles innervated across arbor classes. (**A**) Arbor areas plotted on linear (top) and log₁₀ (bottom) scales for the seven arbor types with arbor areas larger than the surround of a single follicle (i.e. arbor types other than I-FALE, I-FACE, and MCA). (**B**) Axon length vs arbor area. Note the break in the horizontal and vertical axes and the compressions in scale beyond the break. (**C**) Comparisons among the four arbor classes that innervate multiple follicles (HD-FALE, LD-FALE, LA-FACE, SA-FACE): number of follicles innervated vs arbor area, and innervation index (defined as number of innervated follicles per arbor area) vs arbor area. (**D**) Arbor depth within the skin. The skin surface corresponds to 0 μm. On the back skin at P21, the mean depth of a follicle bulb is ~100 μm. (**E**) Orientations of follicle-associated

*Figure 8. Continued on next page*

*Figure 8. Continued*

C-shaped lanceolate endings from 34 LD-FALE arbors relative to the orientation of their associated follicle. In wild type mice at P21, hair follicles on the back are oriented from anterior to posterior, with mean deviations from that axis of less than 10 degrees (**Wang et al., 2010**). Red bars show mean and standard deviation. $V_{NE}$, vector orientation for the nerve ending; $V_{HF}$, vector orientation for the hair follicle.
The following figure supplements are available for figure 8.

**Figure supplement 1**. Differential interference contrast images of parts of BE, LA-FE, LA-FACE, and HD-FALE arbors at different Z-planes.

correlates of this class are, at present, unknown. TE arbors could correspond to the thickened endings of myelinated fibers referred to by Rice and colleagues as 'fuzzy endings' or 'club endings', which are suggested to represent mechanoreceptors (**Fundin et al., 1997**). The two types of arbors with free endings—BE and LA-FE arbors—come close to and may partially penetrate the epidermis (**Figure 8D**), a characteristic that is shared with a variety of peptidergic and nonpeptidergic fibers with free endings, including C-type nociceptors and thermoreceptors. BE arbors bear a particularly close resemblance to the arbors of MrgprB4-expressing neurons, one of a large number of Mrgpr-expressing neurons that together mediate the sense of itch (**Liu et al., 2007**, **2009**).

Tests of these proposed correlations should become possible with methods currently under development. Without using genetic tools, a direct correlation between sensory arbor morphology and single unit physiology is technically challenging, although it has been achieved for neonatal mouse SA I mechanosensory afferents by recording from and then neurobiotin filling individual DRG neurons in an ex vivo skin/DRG/spinal cord preparation (**Woodbury and Koerber, 2007**). As elegantly demonstrated in a study of low threshold mechanosensory neurons, the combined use of this ex vivo preparation and genetically-based cell identification permits both sparse labeling of processes for morphologic analysis and labeling of cell bodies for correlation with electrophysiology (**Li et al., 2011**).

## Functional implications of arbor morphologies

The geometries of sensory arbors make clear predictions about receptive field size and structure. For those I-FALE and I-FACE arbors that target single guard hair follicles, the clear prediction is that sensory information is gathered at single follicle resolution with a sampling pattern based on the regular mosaic of guard hairs. At the opposite extreme, LA-FE neurons are predicted to have receptive fields of 3–5 mm diameter with dense and relatively homogeneous coverage by sensory endings. BE neurons are predicted to have receptive fields of 0.5–1 mm diameter with an even higher density of sensory endings. Spatial resolution is also predicted to be relatively low for HD-FALE arbors, each of which contacts most follicles within its ~0.5 to ~1 mm diameter territory; these afferents presumably sense hair deflections. Many HD-FALE endings cover only part of the circumference of their associated follicle, suggesting that two or three HD-FALE arbors might co-innervate a single follicle and, therefore, that deflection of a single hair might elicit synchronous responses in several HD-FALE arbors.

For any given HD-FALE and LD-FALE arbor, the C-shaped territories of different lanceolate clusters generally reside on the same side of their associated follicles (**Figures 4A–C and 8E**). This asymmetric localization suggests a corresponding asymmetry in mechanosensory responses. By analogy with the responses of auditory and vestibular sensory cells, if hair deflection toward a lanceolate ending ('compression') elicits a response opposite to that of hair deflection away from the lanceolate ending ('extension'), then the asymmetric locations of lanceolate endings would necessarily provide information about the direction of hair movement. A coherent wave of mechanical stimulation—the sort delivered by a large object moving over the body surface—would presumably elicit a response of the same polarity from multiple similarly oriented C-shaped endings. If such a direction-selective response exists it would provide the somatosensory system with an information stream analogous to that provided to the visual system by direction selective retinal ganglion cells (**Vaney et al., 2012**), except that the proposed directionality of the cutaneous sensor would derive from structural asymmetry at the nerve ending/hair follicle complex rather than by spatiotemporal comparisons among neurons.

The pattern of spatially asymmetric C-shaped lanceolate clusters is especially striking among LD-FALE arbors because the innervated follicles are spaced at large intervals (~0.5 mm), a distance that would appear incompatible with any direct communication between endings. If, as seems

probable, the LD-FALE neurons sense coherent hair deflections with low spatial resolution, then these neurons would represent excellent candidates for the C-fiber neurons that subserve affective touch stimuli of the sort elicited by a mother licking her pups (*Andrew, 2010*). Similar C-fiber responses to low force, slowly moving stimuli—referred to as 'pleasant touch'—have also been recorded from human hairy skin by microneurography (*Johansson et al., 1988*; *Löken et al., 2009*).

A pattern of dense follicle coverage is seen in LA-FACE arbors, with ~60% of follicles contacted in a territory of ~2 mm diameter. Although AP histochemistry cannot resolve individual fibers within a tightly packed circumferential bundle, there are clear differences among follicles in the number of AP+ circumferential fibers, since some follicles appear to be encircled by only one or two fibers whereas others exhibit a thick band of AP staining that is at least 10 times the intensity of an individual fiber. If we suppose that each follicle receives roughly the same total number of circumferential fibers, then these data suggest that many follicles receive fibers from two or more LA-FACE arbors, as described above for HD-FALE arbors.

For arbors that contact follicles with lanceolate or circumferential endings, we can estimate the coverage factor—defined as the mean area covered by a particular type of arbor multiplied by the number of arbors of that type per unit area—based on the percentage of follicles contacted within the arbor territory and the patterns of partial occupancy of the contacted follicles. (For the calculations that follow, we remind the reader that all of the data are derived from mice at P21, and it is possible that arbor territories are still being refined at this age.) The average HD-FALE arbor contacts >80% of follicles within its territory and its sensory endings occupy on average >50% of the circumference of each follicle. If we assume that each follicle's circumference is fully and uniformly occupied by lance-loate endings—an assumption supported by high resolution light and electron microscopic imaging (*Casserly et al., 1994*; *Takahashi-Iwanaga, 2000*; *Li et al., 2011*)—then the coverage factor for HD-FALE arbors would be between one and two. This number might be somewhat higher since it appears from the work of *Li et al. (2011)* that lanceolate endings from different arbors interdigitate extensively, with the result that an individual follicle may well accommodate more than two half-circles of lanceolate endings.

A somewhat less precise estimate of the coverage factor can be made for LA-FACE arbors since the extent of coverage of an individual follicle by AP-stained circumferential endings is more difficult to estimate than the extent of coverage by clusters of lanceolate endings. On the assumption that every follicle receives a level of circumferential fiber coverage that roughly corresponds to the highest levels observed in our samples, and with each LA-FACE arbor encircling ~60% of follicles within its territory, we estimate a coverage factor of 2–4. Following the same logic, LD-FALE arbors could potentially have coverage factors of >30 since they innervate hair follicles that are spaced at intervals of ~0.5 mm (*Figure 4A–C*). However, this estimate represents an upper limit, since LD-FALE arbors are presumably competing for follicle targets with HD-FALE arbors. For arbor types without follicle-associated endings, the individual morphologies offer no clues regarding the coverage factor.

## Implications for the developmental biology of cutaneous afferents

The diverse morphologies of cutaneous afferents raise a host of questions regarding developmental mechanisms. In particular, the mechanisms that determine laminar specificity, target specificity, and arbor size and branching density among cutaneous afferents are unknown. While these and related questions have been addressed in a wide variety of neurobiological systems, they may be especially tractable in systems in which arbors are confined to one or several parallel planes, as in the inner plexiform layer of the vertebrate retina and the *Drosophila* larval body wall (*Masland, 2001*, *2011*; *Jan and Jan, 2010*). For sensory arbors that contact multiple follicles, competition among arbors of the same type for a limited pool of targets or target territories—such as the finite circumference of each follicle, around which the lanceolate endings are packed—could determine the size of individual arbor territories. Competition of this sort has been described among lanceolate endings that innervate the same guard hair follicle (*Casserly et al., 1994*), and it is analogous to the competition observed among motor axons at the developing neuromuscular junction (*Turney and Lichtman, 2012*). This model predicts that the average arbor territory will correlate inversely with, and be controlled by, the number of DRG neurons of the given class per unit area of skin. Finally, axon length and number of branch points, which vary over a 1,000-fold range across different arbor types, could be controlled not only by local extrinsic signals but also by regulating, at the level of gene transcription, the abundances and ratios of various cytoskeletal proteins (*Gallo, 2011*).

Another developmental question, hinted at in the preceding section, concerns the mechanism by which C-shaped lanceolate clusters assume a uniformly polarized orientation with respect to the body axes. Hair follicles and their associated structures—Merkel cell clusters (touch domes), arrector pili muscles, and sebaceous glands—are all polarized with respect to the body axes. This is macroscopically apparent in the anterior-to-posterior angle of the hair follicle itself, an organizational feature that is under the control of the planar cell polarity pathway (*Wang et al., 2010*). It seems reasonable to suppose that each hair follicle could transmit polarity information to its associated lanceolate cluster, and recent experiments with mice lacking planar cell polarity signaling in the skin support this conjecture (Chang and Nathans, unpublished observations).

## Skin as a system for studying sensory neuron structure and function

The cutaneous sensory system has a number of favorable experimental attributes that facilitate the integration of experimental approaches. First, the primary sensory neurons are abundant, large, and anatomically segregated (in the DRG) where they can be studied or dissected either during development or in adulthood. Second, the sensory arbors are embedded within a large two-dimensional tissue (the skin) that by virtue of its surface location is readily accessible to controlled stimulation or surgical and pharmacologic manipulations. And third, it is the only sensory system in which single unit recordings can be made from primary sensory neurons in an awake human to correlate physiology with psychophysics (e.g., *Vallbo and Johansson, 1984*; *Vallbo et al., 1984*; *Löken et al., 2009*).

Of the three points noted above, the localization of sensory arbors within the skin is of greatest relevance to the present study. Unlike previous histochemical and immunolocalization studies on skin that have generally been conducted with vertically sectioned tissue (e.g. *Fundin et al., 1997*; *Zylka et al., 2005*), we have worked exclusively with intact skin. This preparation has a number of favorable attributes. First, murine skin is sufficiently thin that intradermal structures can be imaged in their entirety, thus obviating the need for physical sectioning. Second, the two-dimensional nature of the sample and the resiliency of fixed skin simplifies tissue handling, imaging, and cataloguing of structures by their location in the X–Y plane. And third, the relative ease with which large areas of skin can be surveyed in flat mounts facilitates the assembly of large datasets and the identification of relatively rare arbor types.

The present morphologic survey of cutaneous afferents represents one facet of a larger program aimed at systematic structural analyses of nervous tissue. Other efforts include comprehensive descriptions of mammalian retinal cell types and their connectivities; complete reconstruction of the *Drosophila* optic lobe; and light microscope-level reconstruction of mouse neuromuscular units (*Sun et al., 2002*; *Badea and Nathans, 2004*; *Kong et al., 2005*; *Lin and Masland, 2006*; *Takemura et al., 2008*; *Lu et al., 2009*; *Briggman et al., 2011*). Compared to other neuronal reconstruction programs, the analysis of cutaneous sensory arbors has an important simplifying feature: interactions between different arbors are minimal (perhaps limited to co-innervation of the same target follicle) so that a full appreciation of the neuroanatomy can largely be obtained from reconstructions of individual arbors and their epithelial targets. By contrast, in most regions of the nervous system, connectivity between large numbers of pre- and postsynaptic partners is critical to understanding neuronal function. This simplifying feature, together with the technical advantages of two-dimensional imaging and reconstruction in the skin, suggests that, in the not-too-distant future, it should be possible to obtain a comprehensive anatomic description of cutaneous afferents in mammalian skin.

## Materials and methods

### Mice

*Brn3a^CKOAP/CKOAP* mice were crossed to *NFL-IRES-CreER/NFL-IRES-CreER* mice. At GD 17, females carrying *Brn3a^CKOAP/+*; *NFL-IRES-CreER/+* fetuses received 100-500 µg Tamoxifen as an IP injection.

### Histochemistry

Mice were sacrificed at P21, and the back was shaved and treated with hair remover. Back skins were dissected intact and pinned dermal side up to a flat Sylguard surface using insect pins. A series of small cuts at the anterior and posterior ends of each skin served as unique identifiers. Subdermal connective tissue was removed with fine forceps, and the skins were fixed overnight in 4% paraformaldehyde in phosphate buffered saline (PBS), and then heated to 70°C for 90 min to inactivate endogenous phosphatase activity. AP histochemistry and clearing in benzyl benzoate:benzyl alcohol (2:1) (BBBA) were

performed as described in *Badea et al. (2003)*, *(2012)*. For long-term storage, skins were equilibrated in ethanol at −20°C.

## Image analysis

To survey arbor areas, a montage of each skin was first assembled from images obtained with a dissecting microscope. Convex polygons encompassing all isolated (i.e. non-overlapping) arbors were drawn over the image using Adobe Illustrator and each polygon area was calculated. For high resolution analyses, including axon tracing, BBBA-cleared skins were flattened between two glass gel-electrophoresis plates and isolated arbors were imaged at 10× magnification on a Zeiss apotome system. Grey-scale Z-stacks were captured in bright field mode with a 5 μm separation between adjacent layers. Following dehydration in ethanol, with the skin still pinned to the Sylguard surface, and clearing in BBBA, adult skin showed essentially no shrinkage in the X- and Y-dimensions and was reduced to ~92% of the original thickness in the Z-dimension. Therefore ~25 stacks were required to encompass the ~125 μm of full thickness skin in BBBA (original thickness ~135 μm). A montage of images was captured on a Zeiss Imager Z1 and assembled with Zeiss AxioVision software. Neurites were traced using Neuromantic neuronal tracing freeware (Darren Myat, http://www.reading.ac.uk/neuromantic) in semi-automatic mode. As the current version of Neuromantic does not accept input files >1 GB, the files for the largest arbors (the LA-FE arbors), were reduced from ~3 GB to <1 GB by twofold reductions in X and Y resolution. Semi-automated tracing of the larger arbors (LA-FE and LA-FACE) requires ~15 person-hours per arbor. As a measure of reproducibility, we compared two independent traces of the same BE arbor, BE (A11-10) (*Figure 3—figure supplement 1*). The axon lengths and number of branch points for the two tracings were 85,020 vs 86,908 μm and 1011 vs 1071, corresponding to differences of 2.17% and 5.60%, respectively.

The Neuromantic output is a vector file: each line contains a unique number, a set of X, Y, and Z coordinates (the vector's origin), and a second number that indicates the line in the file corresponding to the termination of the vector. The vector files were exported sequentially to Excel (for reformatting), Rotator visualization software (Rotator 3.5), and Adobe Illustrator.

For the analysis of neurite positions at different depths within the skin, complete serial Z-stacks through the skin were obtained with a Zeiss Imager Z1 at intervals of either 2 μm or 5 μm in differential interference contrast (DIC) mode for multiple examples of each arbor type. Neurite positions in the Z-dimension were determined with respect to the epidermal surface and the tips of the hair follicle bulbs.

Statistical analyses were performed with Excel and Graph-Pad. Error bars in the figures indicate standard deviation (SD).

## Acknowledgements

The authors thank Tudor Badea, Hao Chang, Michael Caterina, Xinzhong Dong, David Ginty, and Amir Rattner for helpful advice and/or comments on the manuscript.

## Additional information

### Competing interests

JN: Reviewing Editor, *eLife*, and a paid consultant for Merck; The other authors have declared that no competing interests exist.

### Funding

| Funder | Grant reference number | Author |
| --- | --- | --- |
| Human Frontier Science Program | LT000125/2009-L | Hao Wu |
| Howard Hughes Medical Institute | | Jeremy Nathans |
| The Brain Sciences Institute of the Johns Hopkins University | | Jeremy Nathans |

The funders had no role in study design, data collection and interpretation, or the decision to submit the work for publication.

## Author contributions

HW, Conception and design, Acquisition of data, Analysis and interpretation of data, Drafting or revising the article, Contributed unpublished essential data or reagents; JW, Acquisition of data, Analysis and interpretation of data; JN, Conception and design, Acquisition of data, Analysis and interpretation of data, Drafting or revising the article, Contributed unpublished essential data or reagents.

## Ethics

Animal experimentation: This study was performed in strict accordance with the recommendations in the Guide for the Care and Use of Laboratory Animals of the National Institutes of Health. All of the animals were handled according to approved institutional animal care and use committee (IACUC) protocol MO11M29 of the Johns Hopkins Medical Institutions.

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
