## [Author Response]

*1) Because there are questions about the descriptive nature of the work, it would be helpful if you could take a serious look at the Discussion. We like your writing style, but we think you could strengthen the paper by focusing on a few clear conclusions and avoid diluting the information with speculation that may or may not be correct*.

As suggested here, we have tightened up the Discussion. For example, we have deleted text describing the analogy to the polyspermy block. However, we still like the general idea – perhaps not described with sufficient clarity – that feedback induced by cell–cell contact is a general strategy for producing a winner-takes-all outcome, exactly what is seen during fertilization. Also, the possibility that the arbors are not fully mature at P21 is now noted in the fifth paragraph of this section when we introduce the coverage factor.

*2) There are several caveats about the sampling of sensory neurons labeled in the study. It is not a random sample, but the non-randomness is fully understood by the authors. The NFL Cre driver is not expressed in all sensory neurons. Tamoxifen induction of expression is variable; any low-probability event of this kind is subject to some variation due to factors that are not understood. Brn3a is nearly pan-sensory in embryos but its expression levels change as sensory neurons mature. Only the hairy skin of the back was examined, and so on. These limitations are discussed frankly by the authors and are not an impediment to publication. A truly comprehensive study would require some higher throughput platform to generate and present the data*.

*One issue not fully discussed is the fact that the neurons are hemizygous for Brn3a. Although Brn3a KO heterozygotes are viable and have no gross phenotype, this does not rule out a subtle phenotype in sensory morphology. The strongest argument against this is the known compensatory autoregulation of the Brn3a locus. At least in embryos, this results in near-normal gene expression levels for Brn3a downstream targets in heterozygous knockouts. This is discussed in the cited reference by Eng et al 2004 and this should be noted*.

An excellent point. This is now described at the end of the first paragraph of the Results section, with an additional reference added (39).

*3) The study relies on past descriptions of fiber endings, and the relationship of the endings described here to hair follicles to infer the function of the different endings. However, the present study “likely only covers a fraction of the morphologic diversity of cutaneous arbors”. There are many more morphologies described here than there are known distinct functions. Whether the morphological classification system used here really corresponds well to function remains to be seen. The best support for the validity of the classification method is that the larger arbors seem to have predominantly a single kind of ending, as discussed. This in fact is a significant original contribution, because this was very difficult to appreciate from the kinds of morphological studies in prior papers. The authors observe that LA-FACE arbors sometimes have free endings. One possible reason for this, not mentioned, is that at P21 the arbors are still being refined, and perhaps some of these free endings will either elaborate or regress by adulthood*.

An excellent point. The possibility that P21 arbors are still developing is plausible and is now noted in the second paragraph of the Discussion in the context of the rare “free endings” associated with LA-FACE arbors.